# DNA-Binding Properties of a Novel Crenarchaeal Chromatin-Organizing Protein in *Sulfolobus acidocaldarius*

**DOI:** 10.3390/biom12040524

**Published:** 2022-03-30

**Authors:** Liesbeth Lemmens, Kun Wang, Ebert Ruykens, Van Tinh Nguyen, Ann-Christin Lindås, Ronnie Willaert, Mohea Couturier, Eveline Peeters

**Affiliations:** 1Research Group of Microbiology, Department of Bioengineering Sciences, Vrije Universiteit Brussel, B-1050 Brussels, Belgium; liesbeth.lemmens@vub.be (L.L.); ebert.arno.ruykens@vub.be (E.R.); vantinhnguyen1106@gmail.com (V.T.N.); 2Department of Molecular Biosciences, The Wenner-Gren Institute, Stockholm University, 11597 Stockholm, Sweden; walker359@gmail.com (K.W.); ann.christin.lindas@su.se (A.-C.L.); 3Research Group Structural Biology Brussels, Alliance Research Group VUB-UGent NanoMicrobiology, International Joint Research Group VUB-EFPL NanoBiotechnology & NanoMedicine, Department of Bioengineering Sciences, Vrije Universiteit Brussel, B-1050 Brussels, Belgium; ronnie.willaert@vub.be

**Keywords:** archaea, *Sulfolobus*, nucleoid-associated protein, DNA binding, atomic force microscopy, chromatin structure

## Abstract

In archaeal microorganisms, the compaction and organization of the chromosome into a dynamic but condensed structure is mediated by diverse chromatin-organizing proteins in a lineage-specific manner. While many archaea employ eukaryotic-type histones for nucleoid organization, this is not the case for the crenarchaeal model species *Sulfolobus acidocaldarius* and related species in Sulfolobales, in which the organization appears to be mostly reliant on the action of small basic DNA-binding proteins. There is still a lack of a full understanding of the involved proteins and their functioning. Here, a combination of in vitro and in vivo methodologies is used to study the DNA-binding properties of Sul12a, an uncharacterized small basic protein conserved in several Sulfolobales species displaying a winged helix–turn–helix structural motif and annotated as a transcription factor. Genome-wide chromatin immunoprecipitation and target-specific electrophoretic mobility shift assays demonstrate that Sul12a of *S. acidocaldarius* interacts with DNA in a non-sequence specific manner, while atomic force microscopy imaging of Sul12a–DNA complexes indicate that the protein induces structural effects on the DNA template. Based on these results, and *a contrario* to its initial annotation, it can be concluded that Sul12a is a novel chromatin-organizing protein.

## 1. Introduction

In all domains of life, genomic organization is required in order to pack the chromosome into either a eukaryotic nucleus or a prokaryotic cell and to enable the proper functioning of DNA-based processes such as replication, transcription and recombination. Therefore, folding and compaction of DNA is crucial for organisms to fit their genetic material within the physical boundaries of a cell. In eukaryotes, histone proteins wrap DNA to form nucleosome structures [1], while, in bacteria, nucleoid-associated proteins (NAPs) either locally stabilize long distance contacts by bridging or deform the DNA by bending or wrapping [2].

In archaea, the situation is more complex since the composition of the repertoire of chromatin-organizing proteins differs depending on the considered phylum [3], and the direct impact on the genome organization for each of these proteins largely remains elusive. In Euryarchaeota, proteins similar to eukaryotic histones [4] pack the euryarchaeal nucleoid, forming nucleosome-like structures [5,6] and accelerating DNA mobility [7], whereas in Crenarchaeota chromatin structuring is mainly facilitated by small chromatin proteins that are analogous to bacterial NAPs [8]. Thus far, several NAPs have been described in Crenarchaeota, with Alba proteins being the best characterized [9]. Other characterized crenarchaeal NAPs are small monomeric structurally similar proteins such as Sul7d and Cren7, and Sso10a-family proteins that dimerize in an anti-parallel coiled-coil structure. These NAPs are all characterized by a winged helix–turn–helix (wHTH) structural motif involved in DNA binding [10,11].

When bound onto the double helix, these proteins not only contribute to chromosome organization and compaction but also globally affect gene expression. In Eukaryotes, histone modifications are well-known to modulate chromatin structure and, by consequence, impact gene expression regulation. Surprisingly, even without post-translational modifications, euryarchaeal nucleosomes were shown to repress transcription [12]. Different post-translational modifications, identified on Alba, Sul7d and Cren7, were proposed to have an impact directly or indirectly on gene expression. Indeed, methylation of Sul7d and Cren7 might affect the chromosome structure and, thus, modify transcription [13,14]. In addition, gene expression can be affected by archaeal chromatin proteins (histones and NAPs) due to a competition for DNA binding sites that occurs between these proteins and transcription factors, which are typically gene-specific transcription regulators [7].

Based on the genome annotation of *S. acidocaldarius* [15], a significant number of small DNA-binding proteins are predicted to be encoded in the genome of this crenarchaeal model species. However, there is a limited understanding of which proteins are involved and/or dedicated to chromatin structuring, given that transcription regulators also harbor (w)HTH DNA-binding motifs. Here, we present a study of an uncharacterized small basic protein with a predicted wHTH motif, encoded by the gene *Saci_1012* in *S. acidocaldarius* and named Sul12a (based on its occurrence in Sulfolobales and native molecular weight of 12 kDa). In vitro and in vivo techniques were employed to unravel DNA-binding and -structuring characteristics of the protein. Altogether, is it demonstrated that Sul12a is a small DNA-binding protein that binds DNA in a non-sequence specific manner, thereby seemingly structuring it into a highly condensed structure as is typical for chromatin proteins. By characterizing a novel player in the diverse repertoire of chromatin-organizing proteins, this work contributes to a better understanding of the structuring of the nucleoid in Sulfolobales.

## 2. Materials and Methods

### 2.1. Bioinformatic Analyses

Homology searches were performed using Standard Protein BLAST (National Center for Biotechnology Information) and SyntTax [16]. Clustal Omega version 1.2.4 was used to generate sequence alignments and phylogenetic tree [17]. Archaeal genomes were explored using the UCSC Archaeal Genome Browser [18]. Phyre2 version 2.0 (London, UK) [19], SWISS-Model [20] and AlphaFold version 2.0 (London, UK) [21] were used to predict the protein structure of Sul12a.

### 2.2. Microbial Strains and Growth Conditions

*Sulfolobus acidocaldarius* strain MW001 [22] was cultivated in basic Brock medium [23] supplemented with (*w*/*v*) 0.1% NZ-amine. The pH of the medium was adjusted to 3.5 using sulfuric acid. Given the uracil auxotrophic nature of the strain, the growth medium was supplemented with 10 mg/mL uracil. Microbial growth was performed by incubating at 75 °C in a shaking incubator and was followed by measurement of optical density at 600 nm (OD_600nm_).

*Escherichia coli* strains DH5α and Rosetta DE3 were used for propagation of plasmid DNA and heterologous protein overexpression, respectively. Both strains were grown while shaking at 37 °C in Lysogeny Broth (LB) medium supplemented with 50 µg/mL ampicillin (DH5α) or with 50 µg/mL ampicillin and 34 µg/mL chloramphenicol (Rosetta DE3).

### 2.3. Protein Overexpression and Purification

The *Saci_1012* coding region was amplified by PCR from *S. acidocaldarius* genomic DNA (gDNA) and cloned into a pET-45b(+) expression vector using BamHI and HindIII restriction sites. This resulted in a construct expressing an N-terminally His-tagged recombinant protein.

Heterologous overexpression was accomplished in *E. coli* Rosetta DE3 by allowing the culture to grow until reaching an OD_600nm_ between 0.6 and 0.7, incubating the cells on ice during 30 min and inducing gene expression by the addition of 0.4 mM isopropyl β-D-1-thiogalactopyranoside (IPTG). Subsequently, the culture was further incubated at 37 °C for 16 h. Cells were then harvested by centrifugation, resuspended in lysis buffer (50 mM Na_2_HPO_4_, 0.3 M NaCl, pH 8.0) and lysed by sonication. Lysed cells were centrifuged and the soluble phase containing the heterologously expressed Sul12a protein was collected. Next, the recombinant protein was purified by affinity chromatography using an ÄKTA-fast protein liquid chromatography (FPLC) system with a 5 mL His-Trap FF column (GE Healthcare, Little Chalfont, UK). Fractional elution was accomplished by setting a linear buffer gradient between buffer A (20 mM Na_2_HPO_4_, 0.5 NaCl, 40 mM imidazole, pH 7.4) and buffer B (20 mM Na_2_HPO_4_, 0.5 NaCl, 500 mM imidazole, pH 7.4).

Purity of Sul12a-containing fractions was analyzed by performing 4–12% sodium dodecyl sulfate-polyacrylamide gel electrophoresis (SDS-PAGE). Briefly, 15-μL aliquots were denatured following instructions of the manufacturer of the SDS-PAGE gel electrophoresis system (Thermo Fisher Scientific, Waltham, MA, USA), by adding LDS denaturing buffer (SERVA blue G250, phenol red and lithium dodecyl sulfate at pH 8.5) and heating them at 70 °C for 10 min. This was followed by Coomassie staining. Finally, dialysis into a storage buffer (50 mM Na_2_HPO_4_, 150 mM NaCl, pH 7.4) was performed for fractions containing Sul12a protein in electrophoretically visible purity.

To analyze thermodenaturation of the protein, 50 μL aliquots of crude cell extract containing heterologously overexpressed Sul12a were incubated for 10 min at different temperatures (50 °C, 60 °C, 70 °C, 80 °C, 90 °C and 100 °C), followed by a 2 min centrifugation and SDS-PAGE analysis of 7.5 μL of the supernatant containing non-denatured protein.

### 2.4. Western Blotting

Heterologously purified Sul12a, 100 μg and 250 μg, was separated by SDS-PAGE (4–12%) and then electroblotted onto PVDF membrane (Trans-Blot Turbo from Bio-Rad, Hercules, CA, USA) over 7 min at room temperature in transfer buffer. Membrane was washed with phosphate buffered Saline (PBS)–Tween buffer (10 mM Phosphate pH 7.42, 2.68 mM KCl, 140 mM NaCl, 0.1% Tween 20) and blocked with 5% milk in PBS–Tween for 1 h at 4 °C. Membrane was then probed overnight at 4 °C with 1/1000 horseradish-peroxidase-conjugated anti-polyHistidine antibodies (Sigma-Aldrich, St. Louis, MO, USA), followed by washes with PBS–Tween. Bound antibodies were visualized with ECL mix (Thermo Fisher Scientific Pierce ECL Western Blotting Substrate). The chemiluminescence signals were captured on a CCD camera (Bio-Rad, Hercules, CA, USA) and analyzed with ImageJ package [24].

### 2.5. Size Exclusion Chromatography

Size exclusion chromatography was performed using the ÄKTA-FPLC system with a Superdex 75 column (GE Healthcare, Little Chalfont, UK) equilibrated with 50 mM sodium phosphate buffer (pH 7.4) containing 150 mM NaCl. Calibration of the column was performed by injecting 0.5 mg of each of the following proteins: thyroglobulin (650 kDa), γ-globulin (150 kDa), ovalbumin (44 kDa), myoglobin (17 kDa) and vitamin B12 (1.35 kDa). After injecting 0.5 mg of purified Sul12a protein, the elution volume (Ve) was determined, and the corresponding molecular weight was calculated based on the calibration curve.

### 2.6. Electrophoretic Mobility Shift Assays

To study non-specific DNA binding, an electrophoretic mobility shift assay (EMSA) was performed using linearized plasmid DNA as a binding substrate. Plasmid DNA was purified from a pUC18-containing *E. coli* DH5α culture using a PureYield™ Plasmid Miniprep kit (Promega, Madison, WI, USA) and linearized with restriction enzyme NdeI. Binding reactions were prepared by adding different protein amounts to 110 ng pUC18 DNA in Lrp binding buffer (20 mM Tris-HCl (pH 8.0), 1 mM MgCl_2_, 0.1 mM dithiothreitol (DTT), 12.5% glycerol, 50 mM NaCl, 0.4 mM EDTA) [25] and were incubated at 37 °C during 20 min to equilibrate. Gel electrophoresis of the samples was performed in a 0.8% agarose gel, followed by an incubation within an ethidium bromide containing buffer enabling UV visualization.

For target-specific EMSAs, ^32^P-labelled DNA fragments were prepared by PCR amplification using *S. acidocaldarius* gDNA as a template, a 5′-end-labelled oligonucleotide and a non-labelled oligonucleotide (Appendix A). Labelled oligonucleotides were prepared using [γ-^32^P]-ATP (Perkin Elmer, Zaventem, Belgium) and T4 polynucleotide kinase (Thermo Fisher Scientific, Waltham, MA, USA). Radiolabeled fragments were purified from a native polyacrylamide (6%) gel after electrophoresis. EMSAs were performed as previously described [25] with each binding reaction containing approximately 0.1 nM ^32^P-labelled DNA (corresponding to 12,000 cpm in each binding reaction, determined with a Geiger counter), an excess of non-specific competitor DNA (25 μg/mL sonicated salmon sperm DNA) and varying protein concentrations in Lrp binding buffer. In the case of performing a competitive assay, two different ^32^P-labelled DNA fragments were added at an identical concentration. Binding reactions were incubated at 37 °C during 20 min to equilibrate followed by electrophoresis on native polyacrylamide gels (6%).

Visualization was performed by autoradiography using X-ray sensitive films. The resulting autoradiographs were scanned, and densitometry analysis was performed on these images by measuring the integrated density of the free DNA bands using Image J [24]. The apparent equilibrium dissociation constants K_Dapp_ were then calculated by fitting the density data using a Hill equation as described before [26]. Briefly, the fraction of bound DNA in each lane was calculated as corresponding to the value of 1 with the fraction of unbound DNA substracted. These data were then plotted as a function of the Sul12a protein concentration and fitted using the Hill equation by employing GraphPad Prism software (Appendix A). K_Dapp_ was then determined to correspond to the protein concentration at which the fraction of bound DNA is 0.5.

### 2.7. Chromatin Immunoprecipitation and High-Throughput Sequencing

Chromatin immunoprecipitation (ChIP) was performed as described [27], using anti-Sul12a rabbit antibodies raised against the purified protein (Innovagen, Lund, Sweden) and M-280 Sheep Anti-Rabbit Dynabeads (Invitrogen, Waltham, MA, USA). Mock samples were prepared by incubating an immunoprecipitation reaction with pre-immune serum instead of anti-Sul12a antibodies. Captured gDNA was purified using the iPURE DNA extraction kit (Diagenode, Denville, NJ, USA) according to the manufacturer’s instructions.

ChIP-purified DNA, as well as mock samples and input gDNA were sequenced (1 × 51 bp) by a Miseq sequencer (Illumina, San Diego, CA, USA) at ScilifeLab, Stockholm, Sweden. Sequence reads were mapped to the *S. acidocaldarius* DSM639 genome (NC_007181.1) with a Burrows–Wheeler Aligner (BWA 0.7.10) [28] using default settings and MACS2 (2.1.0) [29] was employed for peak calling. ChIP-seq experiments were performed in biological duplicate and this was followed by manual curation. Finally, ChIP-seq results were visualized by employing IGV version 2.3.59 [30].

### 2.8. Atomic Force Microscopy

Circular supercoiled pUC18 plasmid was purified using a PureYield™ Plasmid Miniprep kit (Promega, Madison, WI, USA) following the provided protocol with two additional wash steps. Protein–DNA complexes were obtained by adding 4.5 ng purified pUC18 plasmid in Lrp binding buffer to varying concentrations of protein in a total volume of 15 μL, followed by a 20 min incubation at 37 °C. After incubation, the prepared samples were mixed with an equal volume of nickel absorption buffer (40 mM HEPES, 10 mM NiCl_2_ (pH 6.74)). For each binding reaction, 10 μL was disposed onto a freshly cleaved mica disc and incubated for another 10 min to allow adsorption of the DNA on the mica. The mica surface was rinsed 5 times with water and gently dried with a stream of air. Atomic force microscopy (AFM) images were obtained using a Multimode Nanoscope IIIa AFM (Bruker, Billerica, MA, USA) operated in tapping mode in air. RTESP AFM probes (Bruker, Billerica, MA, USA) were ozone-cleaned just before use. Images were recorded at a scan rate of 1.5 Hz. Flattening of the images was performed by using NanoScope Analysis v1.5 software (Bruker, Billerica, MA, USA). 3-D images were shown with a pitch of 3°.

## 3. Results and Discussion

### 3.1. Phylogenetic Occurrence of Sul12a

The Sul12a protein is encoded by a 324-bp gene in the *S. acidocaldarius* genome (gene number *Saci_1012*). BLAST analyses indicated that Sul12a is conserved across species belonging to the Crenarchaeal order Sulfolobales, including not only the genera *Sulfolobus* and *Saccharolobus* but also *Metallosphaera* and *Acidianus* (Figure 1a). No homologs were found in other archaeal or bacterial lineages. The *Saci_1012* gene is transcribed as a monocistronic unit in a divergent operon with the other gene encoding a putative ornithine cyclodeaminase. This genetic organization, as well as the neighboring gene synteny are conserved in other Sulfolobales species harboring Sul12a (Figure 1b).

### 3.2. Structural Homology Model and Oligomeric State of Sul12a

The encoded protein in *S. acidocaldarius* is predicted to be a small basic protein, consisting of 107 amino acids (native molecular weight (MW) of 12.4 kDa) and with a predicted theoretical isoelectric point of 8.8. With respect to the other orthologs, conservation is higher in the N-terminal than in the C-terminal domain (Figure 2a), with the N-terminal domain predicted with high confidence to fold into a wHTH DNA-binding motif consisting of three α helices (α1, α2 and α3) and a “wing” defined by β strands β1 and β2 (Figure 2b). The remaining C-terminal 47 amino acids are predicted to adopt an α-helical structure. Depending on the selected model, either two α helices might be formed (α4 and α5; Phyre2 model with highest confidence (Figure 2b)) or a single long α helix is formed (AlphaFold model (Figure 2c)).

The structural homology model reveals similarities with structures of other archaeal chromatin proteins with a wHTH motif, such as Sso10a-2 in *S. solfataricus* [31] and TrmBL2 in *Pyrococcus furiosus* [32], as well as with AbfR2 in *S. acidocaldarius* [33] (Figure 2c). The latter protein has been characterized as a member of the Lrs14 family [34], which is a family of global gene regulators involved in the regulation of biofilm formation and also characterized as a non-specific DNA-binding protein. These structures have in common that the wHTH motif is accompanied by a long amphipathic α helix that mediates dimerization by generating an antiparallel coiled coil structure. As a result, an extended dimeric structure is formed in which the wHTH domains are located at the outer ends of the protein structure, enabling simultaneous interaction with adjacent major groove segments of the DNA. Despite low sequence identities, the similarities between the modelled structure of Sul12a and the structures of the other archaeal non-specific DNA-binding proteins suggests that Sul12a might have a similar DNA-binding mode.

Following heterologous expression of Sul12a, the protein was purified using a His-tag-based affinity chromatography approach resulting in a protein preparation in which a main population of Sul12a protein migrating to its expected MW and lower MW-species were present (Figure 3a). We speculate that these lower-MW species might correspond to proteolytic degradation products of Sul12a in the used buffer conditions. To test this hypothesis, detection of Sul12a using antibodies raised against polyhistidine was performed (Figure 3b). Two main bands were observed: the main one, migrating around 15 kDa, corresponds to the monomeric form of Sul12a, and the second one, migrating around 35 kDa, corresponds to the dimeric form (Figure 3b). Since no lower-MW species were detected using specific antibodies, while they are still observed in denaturing conditions (Figure 3b), they might either be contaminants or proteolytic degradation products of Sul12a, but presenting no tag. To go further, a thermostability analysis, in which cell extracts of an *E. coli* culture heterologously expressing Sul12a were subjected to high temperatures up to 100 °C during 10 min, demonstrated that Sul12a is a highly thermostable protein (Figure 3c). Indeed, while Sul12a protein remains in solution after heat treatment, this is not the case for the *E. coli* proteins, which are removed from the soluble phase after precipitation.

Size exclusion chromatography revealed that Sul12a behaves as a homogenous population of protein species with a predicted MW of 57 kDa (Figure 3d). Given the observation of an extended and non-globular monomeric shape leading to an expected aberrant migration during size exclusion chromatography and the observation of dimeric forms (Figure 3b), we hypothesize that Sul12a forms dimers in solution. This corresponds to most other archaeal chromatin proteins, which are typically monomers or dimers [10,11]. In contrast, TrmBL2 adopts a tetrameric structure with dimer–dimer interactions mediated by the C-terminal domain that is not conserved in Sul12a [32]. To predict the possible oligomerization mode of Sul12a, a structural homology model was made with the SWISS-Model using the N-terminal domain of TrmBL2 as a template given that both proteins have a similar monomeric fold (Figure 3e). This model predicts a dimer with the dimerization helices undergoing antiparallel coiled-coil interactions.

### 3.3. In Vitro DNA-Binding Characteristics of Sul12a

Based on the modeled structure and structural similarities with Sso10a and Lrs14-family proteins, it was hypothesized that Sul12a might interact with DNA non-specifically. Therefore, a DNA-binding assay was performed using linearized plasmid DNA, which demonstrated the formation of slower migrating Sul12a–DNA complexes at protein concentrations of 1 μM and higher (Figure 4). This observation provides evidence that the protein interacts with double stranded (ds) DNA in a non-sequence specific manner.

### 3.4. Genome-Wide Interactions between Sul12a and DNA

As previous assays yielded insights into the DNA-binding mode of Sul12a on ds DNA in vitro, it was then investigated if Sul12a is also interacting in vivo with genomic DNA in *S. acidocaldarius.* To this end, a chromatin immunoprecipitation assay in combination with next-generation sequencing (ChIP-seq) was performed to map genomic interactions using Sul12a-specific polyclonal antibodies. As a negative control, a mock ChIP sample was prepared, to which no antibody was added. This ChIP-seq analysis revealed that there are very little enrichment regions throughout the genome when comparing the immunoprecipitated sample (IP) to the mock control (Figure 5a). Despite this limited enrichment, a set of weaker ChIP-enriched regions were identified (Table 1), however, with only 50 to 150 reads for each peak (Figure 5a).

The identified low-enrichment regions were further tested for in vitro binding by EMSA using purified Sul12a protein and DNA probes each encompassing the summit of one of the ChIP-seq peaks with a length between 300 and 350 bp (Table 1). These experiments verified binding to the enriched regions in vitro as well as binding to a negative control (the promoter region of *Saci_1851*, which was shown not to be enriched in the ChIP-seq analysis) (Figure 5b). Furthermore, an EMSA in which competition in binding was tested between two DNA probes, the ChIP-enriched genomic region nr 9 and a non-related *E. coli* DNA fragment, demonstrated that DNA binding by Sul12a occurs non-sequence-specifically, but is dependent on the length of the used DNA probe (Figure 6). Indeed, while the apparent binding affinity is lower for the enrichment probe as compared to a non-relevant probe when the prior fragment is longer, the opposite is true when the enrichment probe is shorter than the non-relevant probe (Table 2).

The observations of low ChIP enrichment appear in contradiction with the in vitro observations, as Sul12a–DNA interactions are observed in vitro with either short or long DNA probes independent of the sequence (Figure 4 and Figure 5b). Possibly, this inconsistency might be explained by the Sul12a–DNA binding kinetics not enabling a capture in ChIP and/or low intracellular Sul12a concentrations in the tested growth conditions.

### 3.5. Sul12a Affects Architectural Characteristics of the DNA upon Binding

In order to obtain further insights into the binding mode of the protein, AFM imaging was employed to visualize Sul12a–DNA complexes prepared with circular pUC18 plasmid DNA (Figure 7). Several single-molecule images were made for Sul12a concentrations ranging from 7 to 50 bp/Sul12a dimer. Upon comparison to unbound DNA molecules (Figure 7a), the addition of 50 nM Sul12a to the DNA (50 bp/dimer) visually caused conformational changes in the DNA (Figure 7b). Furthermore, it was observed that the protein induced the formation of loops: distant DNA regions within a single plasmid molecule appeared to be co-associated within a single protein-mediated complexed region.

At a higher protein concentration of 130 nM corresponding to 20 bp/dimer, the DNA molecules appeared more compacted with the formation of foci on the DNA, possibly representing condensed DNA in combination with extensive DNA-mediated oligomerization of bound protein (Figure 7c). Large segments of the plasmid molecules that are not associated within these condensed foci appeared not to be bound at all. The formation of foci was not observed in all protein–DNA complexes. Instead, Sul12a appeared to be bound along the whole length of the DNA molecule resulting in bridging events and in some cases the formation of lateral tracks.

At the highest tested Sul12a concentration (250 nM, 7 bp/dimer), extensive compaction of the DNA molecules is observed (Figure 7d). Depending on their appearance, visualized complexes can be divided into following classes: (i) complex structures in which several DNA molecules were linked together by extensively oligomerized protein and leading to the formation of central foci in addition to DNA loops with minimal Sul12a binding; (ii) highly condensed molecular structures apparently each containing a single plasmid molecule, in which in some cases DNA loops were formed. No clear foci were observed in these structures and (iii) complexes in which apparently several DNA molecules were compacted leading to the formation of lateral tracks. Similar DNA-bridging events are also observed for the archaeal chromatin-organizing proteins Alba and Sso10a [12,35] and the bacterial chromatin protein H-NS [36]. These observations underscore the hypothesis that Sul12a is a novel chromatin-organizing protein Sulfolobales and not a specific transcription regulator.

## Figures and Tables

**Figure 1 biomolecules-12-00524-f001:**
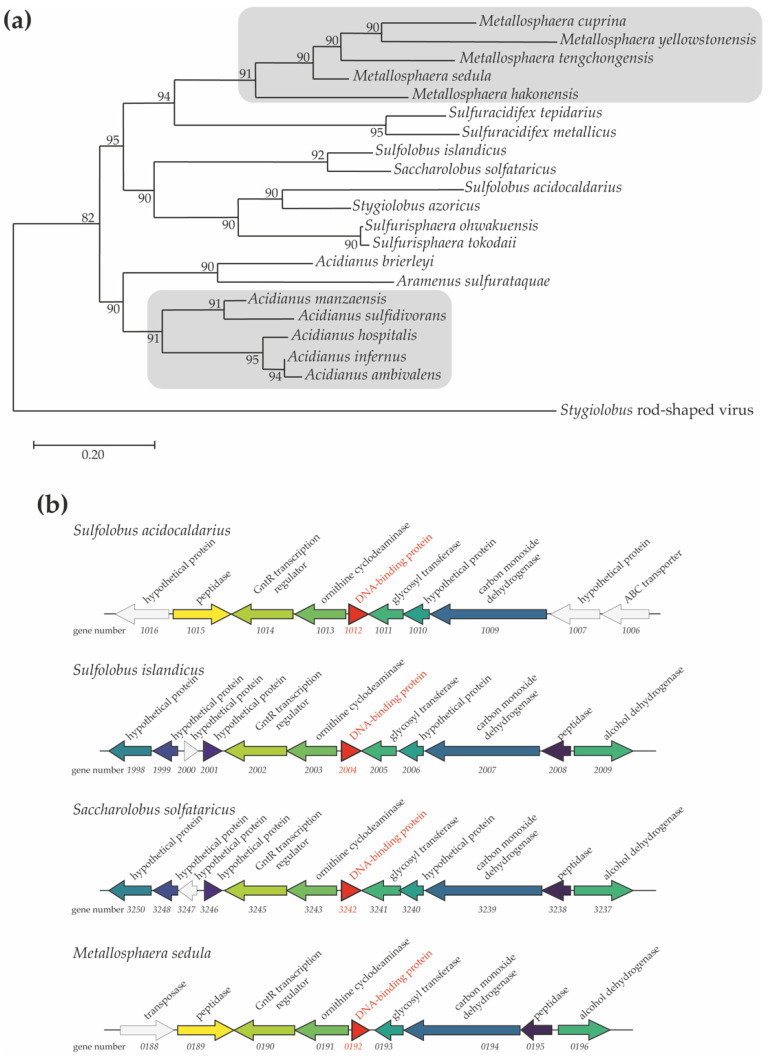
Phylogenetic occurrence of Sul12a. (**a**) Phylogenetic tree of Sul12a homologs based on a clustal omega alignment. The scale bar refers to the number of nucleotide substitutions per site, representing the phylogenetic distance. (**b**) Genome organization of *Sul12a*-like genes (indicated in red as DNA-binding protein) in selected species of Sulfolobales. Predicted gene functions are mentioned based on annotations [15]. Colors refer to orthologous genes.

**Figure 2 biomolecules-12-00524-f002:**
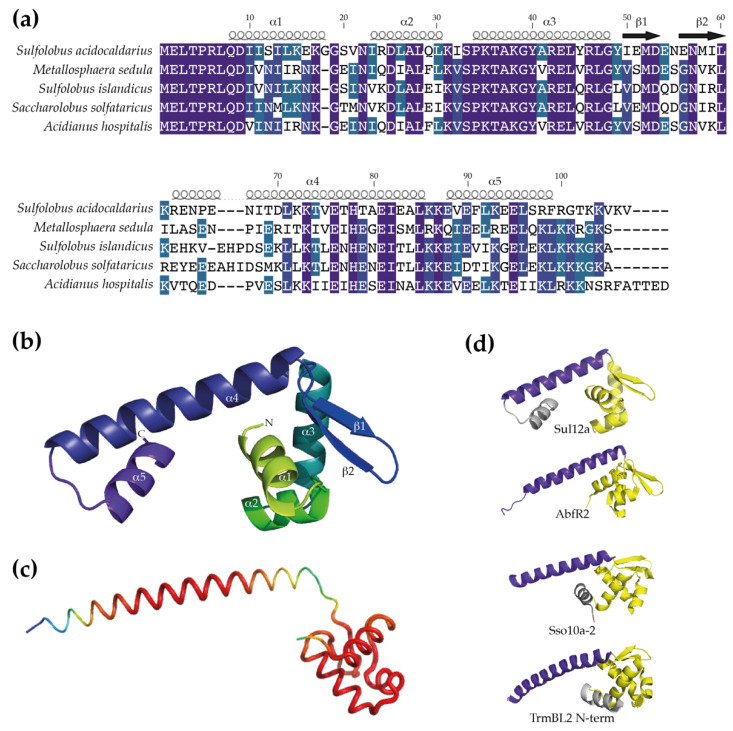
Prediction of the monomeric Sul12a structure. (**a**) Sequence alignment of Sul12a homologs with indication of predicted secondary structure elements. Position numbering is based on the *S. acidocaldarius* Sul12a protein. Pairwise sequence identities and similarities with Sul12a are as follows: 57% and 74% for the *M. sedula* homolog, 59% and 78% for the *S. islandicus* homolog, 58% and 76% for the *S. solfataricus* homolog and 60% and 79% for the *A. hospitalis* homolog. (**b**) Cartoon representation of a homology model of Sul12a, predicted by Phyre2 [19]. This model is based on the crystal structure of the HTH-containing hypothetical protein Sso2273 from *S. solfataricus* (PDB 2X4H) and was selected based on the highest coverage (89%) and confidence score (94.72%) of all models. (**c**) Cartoon representation of a model of the Sul12a structure predicted with AlphaFold [21]. Coloring is according to B-factors, with red depicting higher reliabilities than green and blue. (**d**) Comparison of monomeric structures of Sul12a of *S. acidocaldarius* (model presented in panel (**b**)), Sso10a-2 of *S. solfataricus* (PDB 4HW0), AbfR2 of *S. acidocaldarius* (PDB 6CMV) and the N-terminal domain of TrmBL2 of *P. furiosus* (PDB 5BPI). The core wHTH motif is colored yellow, the dimerization helix purple and additional α-helices are colored grey. The grey α-helix shown for Sso10a-2 is connected to the wHTH motif; this structure is entirely modeled.

**Figure 3 biomolecules-12-00524-f003:**
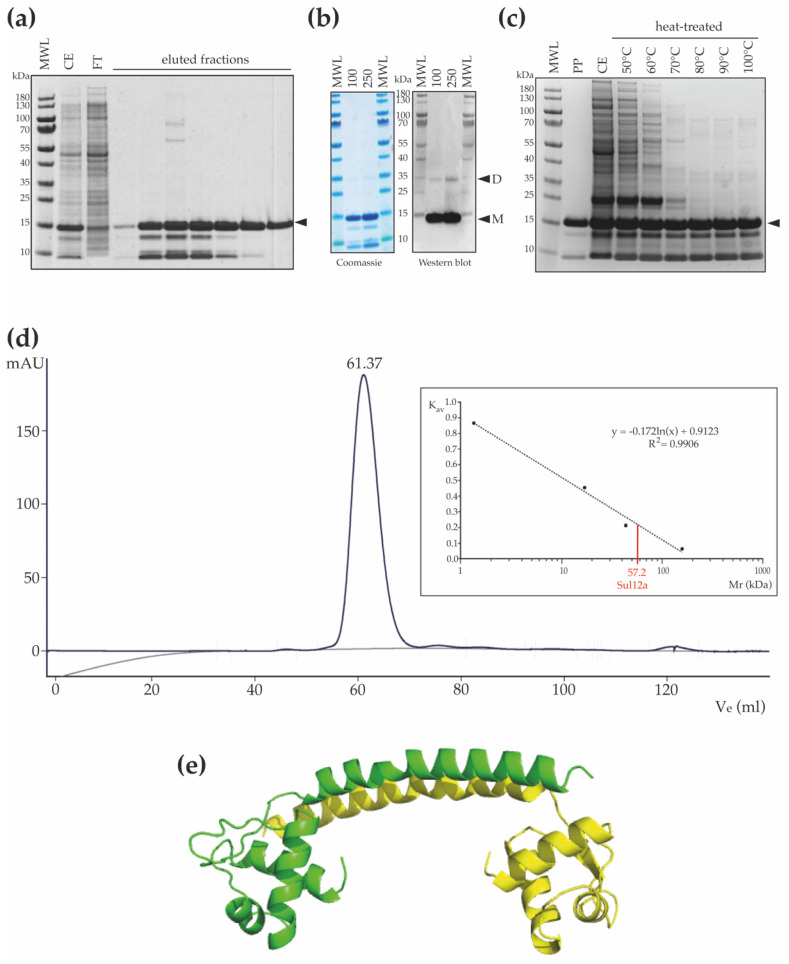
Thermostability and oligomeric characteristics of Sul12a. (**a**) SDS-PAGE of eluted fractions after His-tag affinity chromatography. MWL = molecular weight ladder; CE = crude extract; FT = flowthrough. The monomeric recombinant protein is indicated with a black arrowhead (MW 14.3 kDa). (**b**) SDS-PAGE and western blot of purified Sul12a protein. On the western blot, the molecular weight ladder (MWL) is visible via an aligned overlay. Protein amounts are indicated in μg. “M” stands for a band corresponding to a monomeric state, and “D” stands for a band corresponding to a dimeric state. (**c**) SDS-PAGE of Sul12a-containing crude extracts subjected to a heat treatment at different temperatures. PP = purified protein; CE = crude extract. (**d**) Chromatogram of size exclusion chromatography of recombinantly purified Sul12a protein. The inset displays the calibration curve with indication of the calculated molecular weight of Sul12a based on the measured elution volume (displayed above the elution peak). (**e**) Cartoon representation of a dimeric structural homology model of Sul12a, predicted by the SWISS-Model based on the N-terminal structure of TrmBL2 (PDB 5BPI) as a template. Each monomeric subunit is depicted in a different color.

**Figure 4 biomolecules-12-00524-f004:**
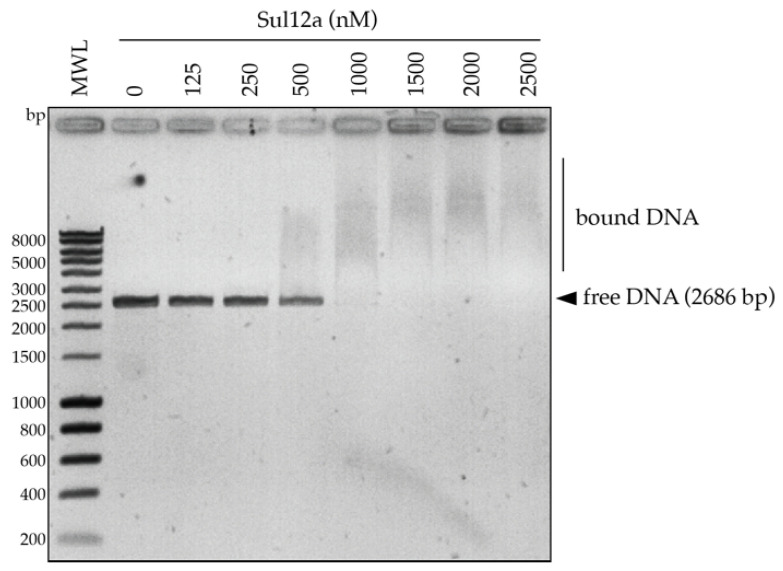
EMSA of Sul12a binding to linearized pUC18 plasmid DNA. Molar protein concentrations are indicated. Populations of free and bound DNA are indicated. MWL = molecular weight ladder.

**Figure 5 biomolecules-12-00524-f005:**
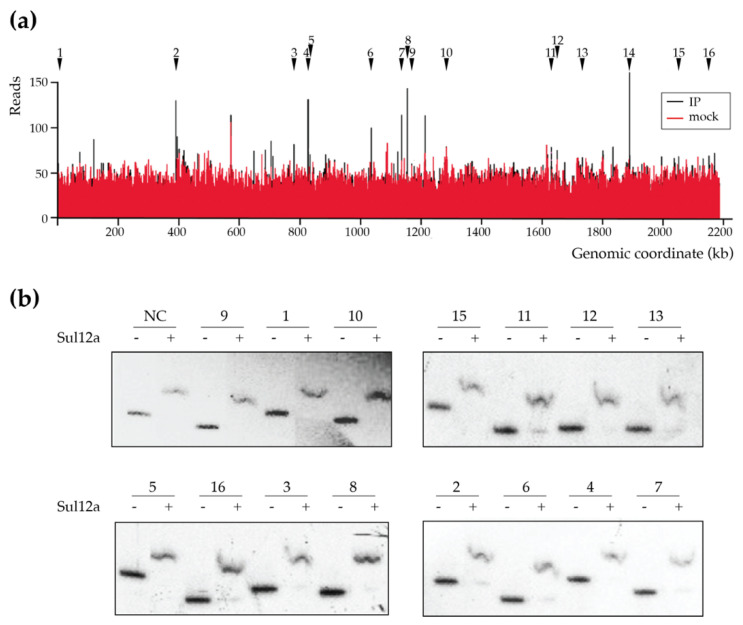
Genome-wide DNA interaction map of Sul12a. (**a**) Overview of the genomic binding profile of Sul12a as monitored by ChIP-seq with indication of numbered selected ChIP-seq peaks that are selected for further analysis (Table 1). IP = immunoprecipitated sample. (**b**) EMSAs of Sul12a binding to radiolabelled DNA probes of 300–350 bp representing the ChIP-seq-enriched genomic regions. A protein concentration of 32 μM was used. Numbering of the probes corresponds to the numbers given to the ChIP-seq-enriched genomic regions in Table 1. NC stands for “negative control”, which corresponds to a 578-bp *carAB* promoter region probe.

**Figure 6 biomolecules-12-00524-f006:**
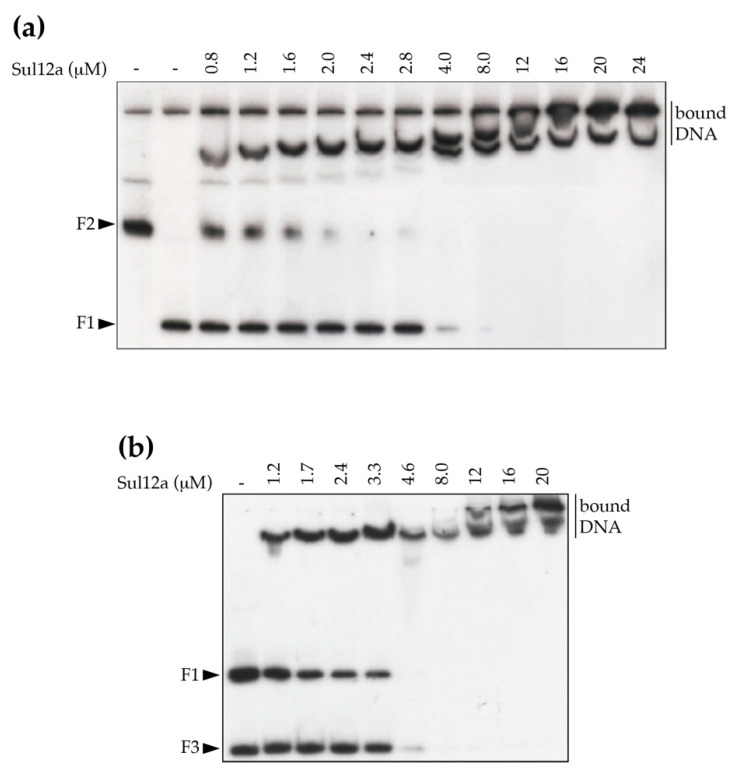
Competition in Sul12a binding for different DNA probes. EMSA were performed in the presence of a mixture of two radiolabeled probes: the ChIP-seq-enriched peak in the neighborhood of the *Saci_1374* gene (F1) and a fragment representing the *E. coli carAB* promoter region (F2) (**a**) or a fragment representing the *E. coli argO* promoter region (F3) (**b**). Molar protein concentrations are indicated.

**Figure 7 biomolecules-12-00524-f007:**
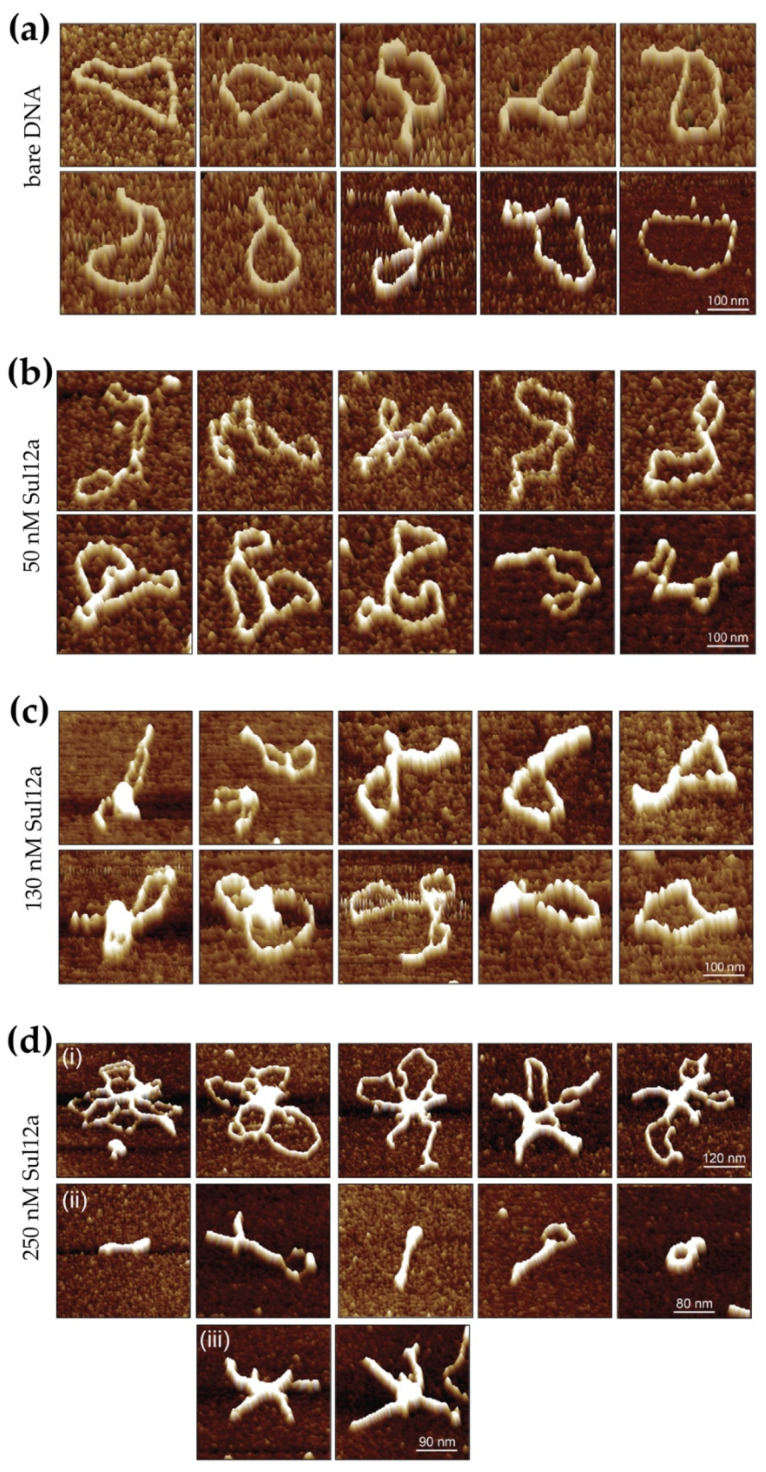
Series of representative AFM height images of individual unbound (bare) pUC18 plasmid DNA molecules and Sul12a–DNA complexes. (**a**) Images of individual molecules of supercoiled bare plasmid DNA (pUC18). (**b**–**d**) Images of individual molecules of Sul12a–DNA complexes, with the protein concentration indicated in dimeric units.

**Table 1 biomolecules-12-00524-t001:** Summary of ChIP-seq results.

Nr	Genomic Coordinates of ChIP-Seq Enrichment Region	Fold Enrichment ^1^	Gene Number of Nearest ORF	Peak Summit Location ^2^
1	11,506–12,946	2.5	*Saci_0017*	G
2	395,203–396,359	4.1	*Saci_0472*	G
3	792,502–793,792	3.4	*Saci_0991*	G
4	840,354–841,874	5.0	*Saci_1041*	G
5	844,058–848,204	2.7	*Saci_1045*	G
6	1,053,261–1,054,510	4.6	*Saci_1237*	I
7	1,142,415–1,144,126	3.2	*Saci_1339*	I
8	1,153,758–1,156,200	3.8	*Saci_1353*	G
9	1,174,705–1,176,888	5.1	*Saci_1374*	G
10	1,283,681–1,285,346	2.6	*Saci_1506*	G
11	1,659,329–1,663,786	2.9	*Saci_1872*	G
12	1,704,818–1,705,948	2.7	*Saci_1906*	G
13	1,758,951–1,762,070	2.8	*Saci_1947*	G
14	1,916,073–1,927,216	6.9	*Saci_2102*	G
15	2,086,681–2,091,395	2.7	*Saci_2246*	G
16	2,189,724–2,193,028	2.5	*Saci_2344*	G

^1^ Average fold enrichment for two replicate experiments. ^2^ It is indicated whether the peak summit is located in an intergenic (I) or intragenic (G) location.

**Table 2 biomolecules-12-00524-t002:** Apparent dissociation constants (K_Dapp_) of Sul12a binding in the competition experiment (Figure 6). Fitted binding curves are shown in (Appendix A).

Probe	Experiment ^1^	Length (bp)	K_Dapp_ (μM)
Enrichment probe 9 (F1)	a	337	3.9
*carAB* promoter region (F2)	a	578	1.2
Enrichment probe 9 (F1)	b	337	2.3
*argO* promoter region (F3)	b	145	3.7

^1^*cfr* panels of Figure 6.

## Data Availability

The ChIP-seq dataset has been submitted to the Gene Expression Omnibus data repository.

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
