# Peer review of "DNA-Binding Properties of a Novel Crenarchaeal Chromatin-Organizing Protein in *Sulfolobus acidocaldarius"

_biomolecules, 2022, doi:10.3390/biom12040524_

Round 1

Reviewer 1 Report

In the manuscript entitled “DNA-binding properties of a novel chromatin-organizing protein in Sulfolobus acidocaldarius”, Lemmens et al describe the identification and characterization of Sul12a, a protein previously annotated as a transcription factor. The work demonstrates ds-DNA biding without sequence specificity as well as structural alterations visualized via AFM. Overall the manuscript is well written and easy to follow.

Specific comments:

Line 48: this sentence is confusing as written

Line 54: the phrase “on the other hand” seems out of place

Line 62: drop the word “the” in the phrase “modify the transcription”

Lines 119-123: did the sample load buffer for SDS-PAGE have SDS? The description of conditions needs more detail.

Line 134: the designation “probe” seems incorrect. Would it be more of a binding substrate?

Line 139: incubation temperature for this assay is abnormally low for a thermophilic protein. With plasmid-length binding substrate, higher temperatures of 50-65C will not appreciably denature the ds-DNA and would better evaluate the binding abilities of the protein. There is a concern that activities of the protein are missed at this low temperature, which is a full 38C below the growth temp of the organism. The manuscript is lacking a temperature activity profile for the protein, so it is unclear as to whether activity is equivalent at 37 and more biologically relevant temperatures. At a minimum, the authors need to justify why 37C is a valid temperature choice in this situation when so many other archaeal biochemistry papers use higher temperatures to evaluate protein:DNA interactions (even for EMSAs).

Line 149: the protein concentrations used need to be specified either in the Methods or in the figure.

Line 150: for a KD determination, the protein concentration needs to be reported. More detail is necessary for how KD was determined using your methodology, including both ligand and receptor concentrations.

Line 151: after the labeled DNA fragments were purified, how was concentration determined to ensure that an identical concentration was used in this step? This is usually very difficult to do with accuracy.

Line 152: why was this assay done at such a low temperature which is not particularly biologically relevant?

Line 157: did you determine cooperativity? This should be inherent in the Hill equation used and is essential information for understanding the potential mechanism behind the compaction behaviour of this protein.

Line 178: same question as for line 152.

Line 202: please report the sequence similarity and identity numbers

Line 228: since the authors have an antibody, the possibility of proteolytic degradation products can be addressed. A western blot should be used to evaluate the cross-hybridization of these putative protein fragments. Additionally, it is known that for some heterologously expressed thermophilic archaeal proteins, there can be forms that migrate faster on an SDS-PAGE gel if not thoroughly denatured by aggressive boiling in SDS. It is possible that the “degradation products” are actually full length, intact protein that has only been partially denatured. The western blot would give some insight here as well.

General protein questions: was this protein tested for ssDNA binding? If so, what was the outcome? If not, why not? Was the protein preparation tested for nuclease activity? What is the site-size for protein:DNA binding?

Line 343: How abundant is this protein intracellularly? If it’s compacting the chromosome in a sequence-independent manner and is involved in “chromatin” formation, it should be relatively abundant. This should be checked by western blot of whole cell extracts.

Figure 5b: What does NC stand for in the figure? How much protein was used for each binding reaction? How much salt is in each reaction and is it matched between input DNA and the bound samples? Addition of purified protein in 150 mM NaCl to the 50 mM NaCl in the buffer would be more salt than in the unbound reactions unless extra salt was added to account for the differential. A higher salt concentration would account for the wavy bands in the figure and knowing how much salt is present would give an indication as to the stability of the protein:DNA interaction (since a salt titration isn’t shown).

Figure 6B: based on the gel, how can the KD for F3 be better than for F1? It’s unclear how this was calculated.

Figure 7: the AFM results are really dramatic and super cool! I love these. For the AFM work, do you have compaction measurements for the DNA? How much is the DNA compressed by the protein? This should be determinable as there are measurement bars in the figure and would be very interesting to know when considering possible compaction models for the chromosome.

Reviewer 2 Report

The article presents in silico work indentifying homolgues of the archaeal protein Sul12a, its structural modelling, the protein purification and in vitro characterization of its DNA binding properties, ChIP, and AFM microscopy of protein-DNA complexes.

It is a very complete and sound work that shows that the protein is a homotetramer, and binds DNA without sequence specificity in vivo by ChIP and in vitro by EMSA. With that in mind the authors suggest that the protein works as a chromatin-organizer, of which there are two other examples in archaea. This is significant because the uniqueness of this chromatin organizers in archaea and the little functional knowledge that exists on them. The article is relevant and the conclusions consistent with the results.

There are a few minor text corrections that would improve the article in my opinion. Please, see attached document.

Reviewer 3 Report

The submitted manuscript used in silico, in vivo, and in vitro methodologies, comprehensively studied the possible structure formation of protein Sul12a, and its potential biochemical properties towards DNA-binding. By these studies, they conclude that Sul12a is a novel chromatin-organizing protein. Although this protein Sul12a may not be of the widest interest for the readers of biomolecules, the research itself is well designed here and problem carefully tackled. Especially, the methodology that was used in this study can be inspiring and motivating for many other biochemical and biophysical studies. I am in support of its publication after considering the following suggestions:

1).  As I mentioned above, one weak point of this manuscript, at least for this journal, is that the studied protein is detailed and limited to one old (and cold) species. Readers may wonder why to care about this protein of Sulfolobus acidocaldarius. Thus, I suggest using Archaea in the title instead of its detailed species name. More meanings and potential impact of this study can be added in the introduction.  

2). The authors used several homology modeling tools such as phyre and swiss-model. Recently AlphaFold2 has shown to be a stunning advance in the protein structure prediction field, outperforming any other classical ways of homology modeling. It is open source and is available online through Colab. I suggest the authors giving a shot there and see how consistent their predictions would be towards classical ways. Related discussions would also be interesting to the general audience.

3). In section 3.2, the authors revealed the oligomeric state of Sul12a. Indeed, it is an important feature for the large-scale packaging of DNA. Previously, the oligomer states of histones and their potential impact on DNA-binding and chromatin packaging were comprehensively studied by Zhao, Papoian, et al. in Biophysical journal 116.10 (2019): 1845-1855. Related work should be cited.

4). The color scheme in Figure 2 (b/c) is not clear. In 2c, Sso10a-2, the grey helix seems separate from the rest. Is it just an angle of view or there is unmodeled region in the middle?

Round 2

Reviewer 1 Report

Thank you for your very clear responses to the queries. You have a very nice bit of work here and it is exciting to see it published!